# Two-site $H_2O_2$ photo-oxidation on haematite photoanodes

Yotam Y. Avital[1], Hen Dotan[2], Dino Klotz [2], Daniel A. Grave[2], Anton Tsyganok[2], Bhavana Gupta[1], Sofia Kolusheva[3], Iris Visoly-Fisher[1], Avner Rothschild [2] & Arik Yochelis [1,4]

$H_2O_2$ is a sacrificial reductant that is often used as a hole scavenger to gain insight into photoanode properties. Here we show a distinct mechanism of $H_2O_2$ photo-oxidation on haematite ($\alpha$-$Fe_2O_3$) photoanodes. We found that the photocurrent voltammograms display non-monotonous behaviour upon varying the $H_2O_2$ concentration, which is not in accord with a linear surface reaction mechanism that involves a single reaction site as in Eley–Rideal reactions. We postulate a nonlinear kinetic mechanism that involves concerted interaction between adions induced by $H_2O_2$ deprotonation in the alkaline solution with adjacent intermediate species of the water photo-oxidation reaction, thereby involving two reaction sites as in Langmuir–Hinshelwood reactions. The devised kinetic model reproduces our main observations and predicts coexistence of two surface reaction paths (bi-stability) in a certain range of potentials and $H_2O_2$ concentrations. This prediction is confirmed experimentally by observing a hysteresis loop in the photocurrent voltammogram measured in the predicted coexistence range.

---

[1] Department of Solar Energy and Environmental Physics, Swiss Institute for Dryland Environmental and Energy Research, Blaustein Institutes for Desert Research (BIDR), Ben-Gurion University of the Negev, 8499000 Midreshet Ben-Gurion, Israel. [2] Department of Materials Science and Engineering, Technion – Israel Institute of Technology, 32000 Haifa, Israel. [3] Ilse Katz Institute for Nanoscale Science and Technology, Ben-Gurion University of the Negev, 8410501 Be'er Sheva, Israel. [4] Department of Physics, Ben-Gurion University of the Negev,  8410501 Be'er Sheva, Israel. Correspondence and requests for materials should be addressed to A.Y. (email: yochelis@bgu.ac.il)

Photoelectrochemical (PEC) water splitting is an elegant route to store radiant solar power in chemical bonds by producing hydrogen[1]. Haematite ($\alpha$-Fe$_2$O$_3$) is an attractive model material for the photoanode in the PEC cell given its visible light absorption[2], abundance, non-toxicity, high catalytic activity for water oxidation[3], and, most importantly, stability in alkaline solutions[4]. Recent advances in haematite photoanodes have led to significant progress in improving their performance, with most of the progress focused on enhancing the photocurrent, reaching over 4 mA cm$^{-2}$ for champion haematite photoanodes[5–8]. The photocurrent records were achieved at electrode potentials of over $1.4\,V_{RHE}$ (Volts vs. the reversible hydrogen electrode)[5–8], which is well above the flat-band potential of haematite photoanodes ($0.3-0.4\,V_{RHE}$)[9]. Enhancing the photocurrent and reducing the applied bias (potential) are required to compete with the efficiency of PV-powered electrolysis systems[10]. Achieving these goals would benefit from detailed understanding of the underlying processes that govern the photogeneration, recombination, and charge transfer at the semiconductor electrode|electrolyte interface[9,11–15].

PEC reactions, such as water splitting and other fuel production reactions, involve three components that give rise to physicochemical processes spanning a wide range of spatiotemporal scales[16–18]. The first component is the semiconductor photoelectrode, where the light absorption, charge carrier generation, separation and transport toward the photoelectrode|electrolyte interface occur. Material properties that control these processes and competing processes that lead to charge recombination play a central role in the photoelectrode operation and performance. The second component is the electrolyte, which supplies the chemical reactants for the fuel production reaction and maintains the ionic conductivity required to close the electrical circuit between the electrodes. The third component is the photoelectrode|electrolyte interface, where the charge transfer reaction occurs (from electronic charges within the photoelectrode to ionic charges within the electrolyte). All three components are cross-linked by feedback loops that follow matter and charge conservation laws.

The oxygen evolution reaction which occurs at the photoanode|electrolyte interface is a complex four-electron process that requires multiple surface intermediates that facilitate interfacial charge transfer with the reactants from the electrolyte[19,20]. On the other hand, those intermediates can also serve as recombination centres for photo-generated charge carriers[21]. Thus, surface intermediates display intriguing and rich dynamics that have a strong influence on the PEC performance of semiconductor electrodes[14,22], which on haematite photoanodes are still under debate[23–25]. Numerous experimental and theoretical reports have suggested the existence of a stable Fe = O intermediate species[24–26]. The existence of this intermediate was recently confirmed by *operando* infrared spectroscopy[26]. The formation of this long-lived intermediate is the first step in the multi-step water photo-oxidation process on haematite photoanodes[26], but it also serves as a centre for recombination of electrons and holes, which results in large overpotential and efficiency losses[14,22].

One way of suppressing the effects of surface recombination in order to study bulk processes within the photoanode is through the use of sacrificial reductants, such as H$_2$O$_2$[27]. At high concentrations (typically about 0.5 M), H$_2$O$_2$ serves as a hole scavenger that collects the photo-generated holes much faster than the surface recombination reaction[28]. Therefore, it provides an upper limit for the bulk-limited photocurrent. Consequently, H$_2$O$_2$ is mostly exploited at high concentrations to gain insights into photoanode properties, such as charge carrier collection and charge transfer efficiencies[27], and it is often considered a surrogate substrate to assess the potential effectiveness of water oxidation co-catalysts[29,30]. However, the mechanism by which H$_2$O$_2$ extracts the photo-generated holes remains elusive. The understanding of this mechanism may lead to rational design of photoanode|electrolyte interfaces and co-catalysts that enhance hole collection and reduce deleterious surface recombination in closely related processes, such as water photo-oxidation.

Here we study the H$_2$O$_2$ photo-oxidation mechanism on a model haematite photoanode. By combining experimental studies at different H$_2$O$_2$ concentrations in alkaline solution together with theoretical modelling of the reaction kinetics, we postulate that the overall H$_2$O$_2$ photo-oxidation reaction, H$_2$O$_2$ + 2 h$^+$ + 2OH$^-$→2H$_2$O + O$_2$, begins with deprotonation of H$_2$O$_2$ to HOO$^-$ in the alkaline solution[31], followed by adsorption of HOO$^-$ ions and interaction with adjacent surface intermediates (Fe = O) of the water photo-oxidation reaction[26]. Thereby, the H$_2$O$_2$ photo-oxidation reaction involves a concerted interaction of two sites, as in Langmuir–Hinshelwood (LH) reactions[32]. The H$_2$O$_2$ photo-oxidation reaction competes with the water photo-oxidation reaction for the holes that arrive at the surface[33], and for the available surface sites and the intermediate species that facilitate both reactions. The consequences of this competition are unfolded by a nonlinear kinetic model that not only reproduced the experimental observations but also gave rise to unexpected predictions that were subsequently verified by further experiments. Unlike the linear models of water photo-oxidation on haematite that consider a multi-step reaction on a single surface site[24–26,34,35], as in Eley–Rideal (ER) reactions, our work suggests that splitting the reaction into two sites, as in LH reactions, enabled here by the presence of H$_2$O$_2$ in the electrolyte, gives rise to nonlinear behaviour and facilitates the collection of photo-generated holes by oxidized surface species that help to level the potential of the elementary steps involved in the reaction[20]. The knowledge gained from this work may explain the behaviour observed for some common water oxidation co-catalysts based on cobalt[36–38], and be also used to rationally design co-catalysts that facilitate water oxidation via a two-site process[20].

In what follows, we first discuss photocurrent vs. potential voltammetry measurements and intensity-modulated photocurrent spectroscopy (IMPS) measurements carried out at different H$_2$O$_2$ concentrations ranging from 0 to 0.5 M in alkaline solution (1 M NaOH in deionized water). Secondly, we model and analyse the underlying surface reactions and calculate the photocurrent as a function of potential and H$_2$O$_2$ concentration. Thirdly, we show that the calculated results qualitatively reproduce the empirical observations and finally, we verify experimentally the model prediction that is related to hysteresis.

## Results

**PEC measurements**. The photocurrent ($j$) vs. potential ($U$) voltammograms are shown in Fig. 1a; the measurements were done under white light-emitting diode (LED) illumination of 80 mW cm$^{-2}$ (see Methods section for details) at H$_2$O$_2$ concentrations ranging from 0 to 500 mM in alkaline solution (1 M NaOH in deionized water). The photocurrent density, $j$, is the current measured under illumination after subtraction of the dark current (at the same potential), divided by the aperture area of the PEC cell. At low H$_2$O$_2$ concentrations (5 mM), the photocurrent voltammograms display three regions distinguished by their slope, i.e., weak increase in $j$ with increased $U$ up to point (i), rapid increase from point (i) to point (ii), and moderate increase from point (ii) onwards. At higher H$_2$O$_2$ concentrations (10 mM) $j$ increases almost linearly with $U$, indicating that the flux of photo-generated holes arriving at the surface is proportional to

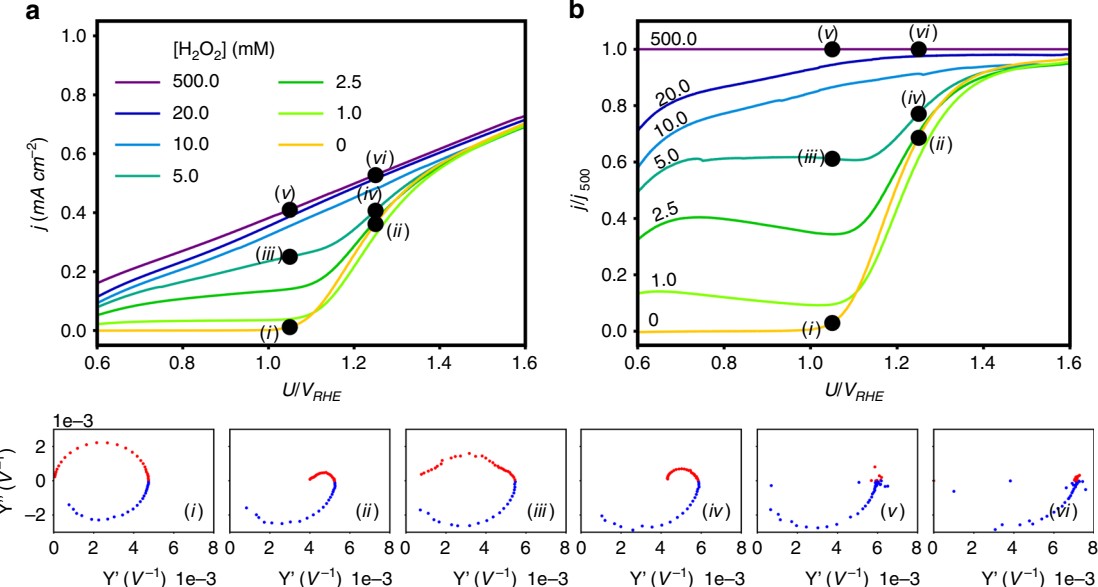

**Fig. 1** Photocurrent voltammograms and intensity-modulated photocurrent spectra at varied $H_2O_2$ concentrations. **a** Photocurrent density as a function of the applied potential under white LED illumination (80 mW cm$^{-2}$) at different $H_2O_2$ concentrations as indicated in the legend. **b** Normalized photocurrent voltammograms obtained by dividing the photocurrent voltammograms in **a** by the photocurrent measured at [$H_2O_2$]= 500 mM, $j_{500}$. Bottom IMPS spectra correspond to: (i) [$H_2O_2$]=0 mM, $U$= 1.05 V$_{RHE}$, (ii) [$H_2O_2$]=0 mM, $U$= 1.25 V$_{RHE}$, (iii) [$H_2O_2$]= 5 mM, $U$= 1.05 V$_{RHE}$, (iv) [$H_2O_2$]= 5 mM, $U$= 1.25 V$_{RHE}$, (v) [$H_2O_2$]= 500 mM, $U$= 1.05 V$_{RHE}$, (vi) [$H_2O_2$]= 500 mM, $U$=1.25 V$_{Rhe}$

the applied potential. To distinguish the bias-dependent interfacial charge transfer at the photoanode|electrolyte interface from the bias-dependent hole flux from the bulk to the surface, the photocurrent voltammograms were normalized by the photocurrent obtained at the highest $H_2O_2$ concentration (500 mM), for which the surface reaction no longer limits the photocurrent[27]. The normalized photocurrent voltammograms are depicted in Fig. 1b. They display an unexpected non-monotonous behaviour at low $H_2O_2$ concentrations (from 1 to 5 mM) wherein the normalized photocurrent first increases at low potentials, secondly decreases at moderate potentials, thirdly exhibits a steep rise at the onset potential around 1.1 V$_{RHE}$, and finally converges with the water photo-oxidation current at about 1.5 V$_{RHE}$.

To shed more light into the photocurrent behaviour at different $H_2O_2$ concentrations, IMPS measurements were carried out at potentials and $H_2O_2$ concentrations marked by points (i) to (vi) in Fig. 1a, b. The IMPS spectra of the complex photocurrent admittance, $Y(\omega) = j(\omega)/\Phi(\omega)$, where $\omega$ is the ac frequency and $\Phi$ is the light intensity, display two main features: upper (positive imaginary part) and lower (negative imaginary part) semicircles. The diameter of a lower semicircle corresponds to hole flux to the surface, whereas the diameter of the upper semicircle corresponds to surface recombination[39]. In the absence of $H_2O_2$ in the electrolyte, the upper semicircle at low potentials (point (i)) is large, indicating strong surface recombination. Notably, the $Y(\omega)$ vanishes when $\omega \to 0$, indicating that all photo-generated holes arriving at the surface recombine with electrons such that an increase in light intensity does not lead to an increase in photocurrent. At higher potentials (point (ii)), the upper semicircle shrinks, indicating lower surface recombination as usually observed in haematite photoanodes in alkaline solution (without $H_2O_2$) at high potentials[39]. The spectra obtained at a high $H_2O_2$ concentration (500 mM, points (v) and (vi)), display only the lower semicircle, whereas the upper semicircle disappears both at low and high potentials. This indicates that at high concentrations, $H_2O_2$ serves as an effective hole scavenger that suppresses surface recombination as expected[27]. However, at

a low $H_2O_2$ concentration (5 mM, points (iii) and (iv)) surface recombination is strong at low potentials (point (iii)) and is suppressed at higher potentials (point (iv)), similarly to the situation without $H_2O_2$ (points (i) and (ii), respectively). These analogies indicate that in points (iii) and (iv), $Y(\omega)$ is dominated by water photo-oxidation rather than by $H_2O_2$ photo-oxidation. This implies that at low $H_2O_2$ concentrations, the photocurrent below the onset of water photo-oxidation is limited by the supply of $H_2O_2$ to the surface.

The transition from $H_2O_2$-limited photocurrent at low potentials to $H_2O_2$-independent photocurrent at high potentials implies a competition between $H_2O_2$ photo-oxidation and water photo-oxidation for the available holes, surface reaction sites and intermediate species[33]. To test this hypothesis, we examine the effect of light intensity, which controls the photogeneration rate of excess charge carriers (holes), at a low $H_2O_2$ concentration of 2.5 mM. The results are shown in Fig. 2. The pronounced nonlinear behaviour observed at high light intensities becomes linear as the light intensity is decreased. This shows that at low light intensity, the photo-generation rate is low enough to enable hole scavenging even at low $H_2O_2$ concentrations. Another observation that supports our hypothesis is the saturation of the photocurrent density with increasing light intensities at potentials lower than 1.1 V$_{RHE}$, which indicates that the photocurrent is limited by $H_2O_2$ rather than by the flux of photo-generated holes.

**Modelling of the reaction kinetics.** The experimental results presented in the previous section suggest that the fate of the photo-generated holes arriving from the bulk (where they are created) to the surface of the photoanode (where they vanish) involves different possible paths: $H_2O_2$ photo-oxidation, water photo-oxidation and recombination with electrons (surface recombination). The competition between these paths depends on control parameters such as the $H_2O_2$ concentration, light intensity and applied potential, thereby affecting the

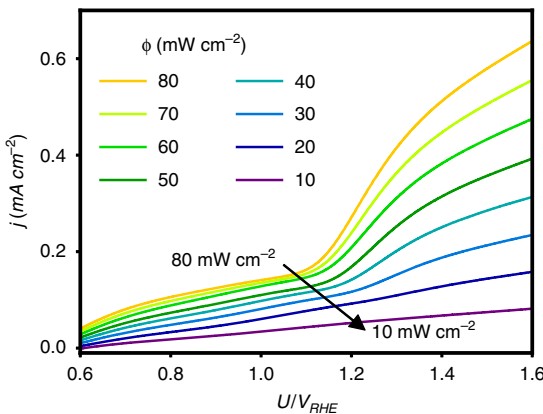

**Fig. 2** Photocurrent voltammograms under different illumination intensities. For all measurements $H_2O_2$ concentration is at 2.5 mM

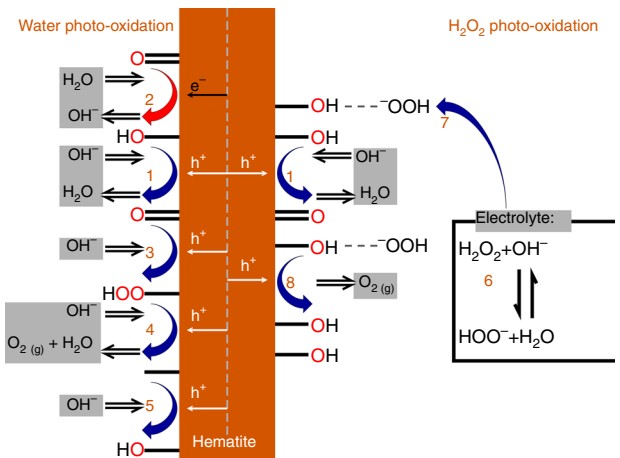

**Fig. 3** Schematic illustration of water (left) and $H_2O_2$ (right) photo-oxidation reaction paths in alkaline solution. The former follows steps (1) to (5), whereas the latter follows steps (6) to (8) combined with step (1). In addition, the red arrow denotes the recombination reaction (2)

photocurrent. In order to understand the mechanisms of the respective reactions and the competition between them that determines the fate of the photo-generated holes, and thereby the performance of the photoanode, we construct a kinetic model. Our model extends the water photo-oxidation mechanism presented by Zandi and Hamann[26] and Zandi and co-workers[40], by adding additional steps that account for $H_2O_2$ photo-oxidation. The model yields kinetic equations, which we solve numerically in order to analyse the fate of the photo-generated holes as a function of the control parameters, aiming for qualitative agreement between the calculations and the experimental results in the entire parameter space examined. Since we aim for qualitative rather than quantitative agreement, our model is not so sensitive to the selection of the parameters (rate constants) in the kinetic equations, unlike multi-parametric fitting that aims for quantitative agreement by fitting the unknown parameters to reproduce the experimental results precisely. Instead, the essence of our model is in capturing the nonlinearity of the PEC kinetics that gives rise to the non-trivial behaviour observed in the experimental results presented in Figs. 1, 2. Therefore, the model depends on the phenomenological structure of the nonlinear kinetic equations rather than on the parameters within those equations.

We start with the widely accepted model of water photo-oxidation mechanism on haematite in alkaline solution[26,40,41]:

$$\boxed{\text{Fe-OH}} + \text{OH}^- + h^+ \xrightarrow{k_1} H_2O + \boxed{\text{Fe}=\text{O}}, \tag{1}$$

$$\boxed{\text{Fe}=\text{O}} + H_2O + e^- \xrightarrow{k_{-1}} \boxed{\text{Fe-OH}} + \text{OH}^-, \tag{2}$$

$$\boxed{\text{Fe}=\text{O}} + \text{OH}^- + h^+ \xrightarrow{k_2} \boxed{\text{Fe-OOH}}, \tag{3}$$

$$\boxed{\text{Fe-OOH}} + \text{OH}^- + h^+ \xrightarrow{k_3} O_2 + H_2O + \boxed{\text{Fe}}, \tag{4}$$

$$\boxed{\text{Fe}} + \text{OH}^- + h^+ \xrightarrow{k_4} \boxed{\text{Fe-OH}}, \tag{5}$$

where boxed ($\boxed{\cdots}$) species correspond to surface intermediates and $k_i$ are the respective rate constants. The reaction path from steps (1) to (5) is illustrated on the left side of Fig. 3. According to this mechanism, the water photo-oxidation current is given by the net summation of all the reaction rates, that is, the sum of the forward charge transfer rates (steps (1) and (3)–(5)) minus the backward reaction rate (step (2)). When both reactions (1) and (2) occur successively, their combination forms effectively a

surface recombination step. It is noteworthy that steps (1)–(5) give rise to a set of linear kinetic equations, see Model equations and numerical computations in Methods.

Next, incorporating the $H_2O_2$ photo-oxidation reaction requires accounting for the following observations: (a) At low and intermediate potentials, the photocurrent depends on the $H_2O_2$ concentration, whereas at high potentials it coincides with the water photo-oxidation current (without $H_2O_2$), see Fig. 1a, b. This suggests that the $H_2O_2$ photo-oxidation reaction competes with the water photo-oxidation reaction[33]. (b) Comparison of the IMPS spectra in points (i) and (iii) in Fig. 1 shows a deviation of the upper semicircle from a perfect shape in the presence of $H_2O_2$ in the electrolyte, which suggests that the $H_2O_2$ photo-oxidation reaction competes with the recombination reaction (step (2)). (c) Since the stable surface intermediate in the water photo-oxidation reaction is Fe = O[26,40,41], it is expected that the competing step in the $H_2O_2$ photo-oxidation reaction should involve Fe = O intermediates. (d) In the presence of $H_2O_2$ in the electrolyte the photocurrent at low potentials is due to $H_2O_2$ photo-oxidation rather than water photo-oxidation. Under these conditions, the Fe = O intermediates are short-lived and the stable surface species is Fe-OH[26]. Thus, for $H_2O_2$ photo-oxidation to occur, an adsorption step that involves the long-lived Fe-OH species is expected to precede the step in which the adions interact with the Fe = O intermediates, as suggested in (c).

Accordingly, we postulate that the water photo-oxidation reaction (steps (1)–(5)) is complemented by the following steps to account for the $H_2O_2$ photo-oxidation reaction:

$$H_2O_2 + \text{OH}^- \rightarrow \text{HOO}^- + H_2O, \tag{6}$$

$$\text{HOO}^- + \boxed{\text{Fe-OH}} \xrightarrow{k_5} \boxed{\text{Fe-OH} \cdots ^-\text{OOH}}, \tag{7}$$

$$\boxed{\text{Fe}=\text{O}} + \boxed{\text{Fe-OH} \cdots ^-\text{OOH}} + h^+ \xrightarrow{k_6} 2\boxed{\text{Fe-OH}} + O_2, \tag{8}$$

Step (6) describes the deprotonation reaction of $H_2O_2$ in the alkaline solution[31], which occurs spontaneously since a p$K_a$ of 11.7 for $H_2O_2$ deprotonation[42] yields $[\text{HOO}^-]/[H_2O_2] \simeq 100$ at pH = 13.6. Due to the fast nature of deprotonation reactions, step (6) is expected to occur much faster than the other steps, thus it does not affect the overall kinetics. Its product, $\text{HOO}^-$, is postulated to weakly (physically) adsorb to Fe-OH surface sites in

the next step, as described in step (7), in accordance with (d) above. Empirical evidence supporting this adsorption step was obtained by infrared spectroscopy (see Supplementary Figure 1 and related discussion in the Supplementary Information). It is also supported by previous works showing the adsorption of (non-deprotonated) $H_2O_2$ to Fe-OH surface sites in haematite[43] (and similarly in other transition metal oxides[44]). Finally, step (8) postulates that the adsorbates from step (7) are oxidized by photo-generated holes coupled with proton transfer to adjacent Fe = O surface intermediates that were produced in step (1) of the water photo-oxidation reaction. Thus, step (8) represents a concerted interaction between two different surface species, one from the water photo-oxidation reaction and the second from $H_2O_2$ adsorption. It couples both reactions and gives rise to nonlinearity in the kinetic equations that depend on the concentration of the respective species (see Model equations and numerical computations in Methods for details). It is noted that the exact molecular identity of the surface species involved in the reaction is beyond the scope of this work, and it remains to be verified by spectroelectrochemical studies[45]. However, this specific detail is not crucial to the results that follow, and alternative surface species may be considered without affecting the qualitative results of the analysis. The salient point is that the $H_2O_2$ photo-oxidation reaction mechanism involves a concerted interaction of two surface sites, as in LH reactions, consuming two holes and yielding an oxygen molecule. The postulated reaction path is schematically illustrated on the right side of Fig. 3.

To uncover the nonlinear nature of the elementary steps in the water and $H_2O_2$ photo-oxidation reactions that result in the measured non-monotonic photocurrents for certain $H_2O_2$ concentrations (see Fig. 1), we derive kinetic equations (see Model equations and numerical computations in Methods for details) and supplement them by the hole flux from the surface to the electrolyte that is given by the sum of the forward chemical reactions (1), (3)–(5), and (8):

$$k_1 p_4 \theta_{\text{Fe−OH}} \sigma_{\text{h}} + k_2 p_4 \theta_{\text{Fe}=\text{O}} \sigma_{\text{h}} + k_3 p_4 \theta_{\text{Fe−OOH}} \sigma_{\text{h}} + k_4 p_4 \theta_{\text{Fe}} \sigma_{\text{h}}$$
$$+ k_6 \theta_{\text{Fe−OH···OOH}} \theta_{\text{Fe}=\text{O}} \sigma_{\text{h}} = p_1,$$
$$(9)$$

where $p_1$ is the hole flux from the surface to the electrolyte[46], which in general depends on the illumination intensity and potential, and $p_4$ is the $OH^-$ concentration in the electrolyte, which depends on the electrolyte composition, $\theta_x$ is the fractional surface coverage of species $x$, and $\sigma_{\text{h}}$ corresponds to the surface

density of holes at reacting sites. It is useful to regard eq. 9 as the charge conservation constraint[46]. In addition to the charge conservation constraint, we also employ a standard surface site conservation constraint:

$$\theta_{\text{Fe−OH}} + \theta_{\text{Fe}=\text{O}} + \theta_{\text{Fe−OOH}} + \theta_{\text{Fe}} + \theta_{\text{Fe−OH···OOH}} = 1. \quad (10)$$

Next, we define the normalized photocurrent that is related to the flux of holes across the surface ($p_1$) minus the flux of holes consumed by the surface recombination reaction (step (2)):[46]

$$\frac{j}{j_{\text{max}}} = 1 - \frac{k_{-1} \theta_{\text{Fe}=\text{O}} p_2}{p_1}. \quad (11)$$

where $p_2$ is the electron density at the surface. We emphasize that this form regards only the reaction kinetics at the surface and it does not account for bulk processes that control the hole flux to the surface. From experiments, it is evident that $p_1$ is often potential dependent, but the dependence is relatively weak. For instance, in the results presented in Fig. 1 the dependence is essentially linear (see $j_{500}$ in Fig. 1). In contrast to $p_1$, $p_2$ displays strong dependence on the applied potential, $p_2 \propto \exp(-\varphi)$, where $\varphi$ is dimensionless potential (see Model equations and numerical computations in Methods section for details). Thus Eq. (11) comprises an empirical normalization by $p_1$ (similarly to the normalization by $j_{500}$ in Fig. 1b). Notably, the shape of the calculated voltammograms of the normalized photocurrent is essentially robust with respect to the potential dependence of $p_1$, as shown here by comparing Fig. 4a, b in which $p_1$ is taken to be linearly proportional to $U$ or independent of $U$, respectively. This implies a generic mechanism that is hidden in this physicochemical process that we unfold, in what follows, by using a bifurcation theory via keeping $p_1$ as constant for simplified analysis purposes, $p_1 = p_1^{\text{c}}$.

The qualitative similarity between the calculated curves in Fig. 4 and the experimental results in Fig. 1b is clearly evident. Specifically, the calculated curves capture the non-monotonous behaviour of the normalized photocurrent at low $H_2O_2$ concentrations and potentials below the onset of water photo-oxidation, and the convergence of the photocurrent at high potentials towards the water photo-oxidation limit. It is noted that the non-monotonous behaviour displayed by the cyan ($p_3 = 0.01$) curve in Fig. 4 emerges from the nonlinearity in the kinetic equations, as a result of the concerted interaction of two different surface sites as in step (8) above.

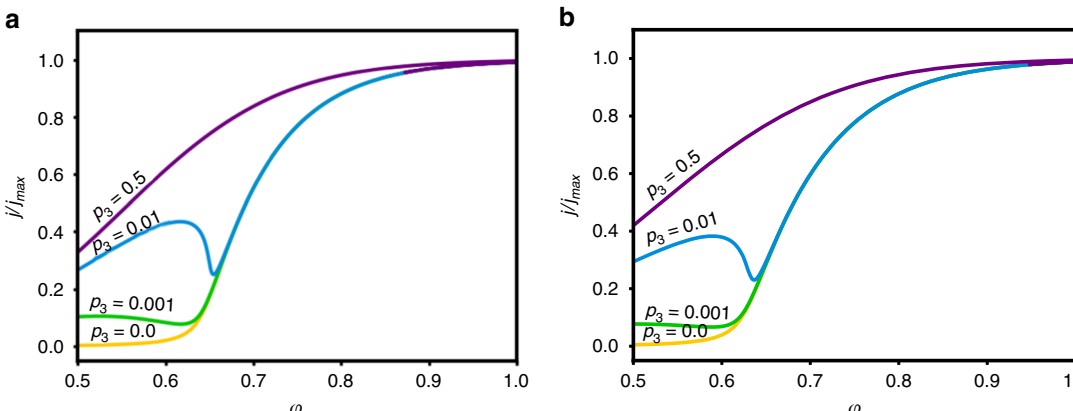

**Fig. 4** Numerically computed normalized photocurrent voltammograms at different $H_2O_2$ concentrations ($p_3$). All the observables are dimensionless. The photocurrents are calculated for two cases in which $p_1$ is **a** with linear dependence on the applied potential, or **b** constant (see text and Model equations and numerical computations in Methods section for details)

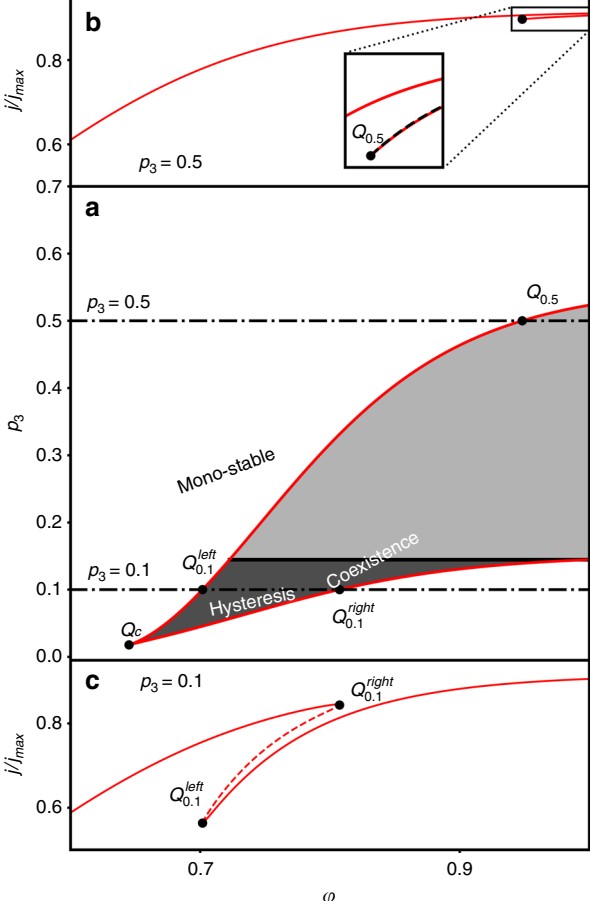

**Fig. 5** Bi-stability and hysteresis. **a** Coexistence parameter space for the computed normalized photocurrent voltammograms showing the distinct regions of coexisting solutions. 2D projection of the manifold describing the computed normalized photocurrent voltammograms ($j/j_{max}$) as a function of the applied dimensionless potential ($\varphi$) and $H_2O_2$ concentration ($p_3$), showing the coexistence (bi-stability) regime of the normalized photocurrent (the grey shaded area). The bold dashed line indicates the threshold between observable and non-observable hysteresis. **b**, **c** Simulated normalized photocurrent voltammograms at $p_3 = 0.5$ and 0.1, respectively. Solid lines indicate stable solutions, whereas dashed lines indicate unstable ones. The transition points from stable to unstable solutions and vice versa in **b**, **c** are, respectively, marked in **a**

**Bi-stability, competing kinetics and hysteresis.** Analysis of the solutions shows that while for low $H_2O_2$ concentrations there is only one solution (mono-stable regime), above a certain threshold concentration there is a limited potential range in which stable solutions coexist, as shown in Fig. 5a, c. The coexistence of stable solutions implies that hysteresis in the photocurrent should be expected in this range of potentials and $H_2O_2$ concentrations. The hysteresis onset is at a generic cusp bifurcation[47] (marked by $Q_c$ in Fig. 5a) in the two-parameter projection of the normalized photocurrent as a function of the potential ($\varphi$) and $H_2O_2$ concentration ($p_3$), see Fig. 5a. The shaded region between the curves in Fig. 5a is where coexistence of bi-stable solutions exists. Above $p_3 \simeq 0.144$ (marked by the bold dashed line in Fig. 5a), although the coexistence persists, it cannot be observed experimentally because one of the stable solution branches spans the entire relevant potential range. In this case, an unstable solution connects the two stable solutions only at infinite potential, a known and universal (model independent) property of the cusp bifurcation[47]. Thus, our calculations predict a stable unique behaviour

at low and high $H_2O_2$ concentrations, whereas at intermediate concentrations bi-stable solutions and hysteresis are expected (in a certain potential range). It is noted that additional calculations (not shown here) demonstrate the same qualitative behaviour with similar features of hysteresis and multiplicity of solutions (i.e., bi-stable regimes) over a wide range of parameters spanning orders of magnitude of the respective rate constants, confirming the robustness of our model predictions. Moreover, the simple ER mechanism typically used to describe water photo-oxidation on haematite photoanodes[24,25,34,35] cannot yield such hysteresis because it lacks nonlinearity in the kinetic equations of its elementary steps (see Model equations and numerical computations in Methods for details).

The hysteresis predicted by the model calculations was validated by linear sweep voltammetry measurements, as shown in Fig. 6. Consistent with the model predictions, hysteresis was observed only at intermediate $H_2O_2$ concentrations (5 mM; Fig. 6b) but not at low and high concentrations (Fig. 6a, c, respectively). The hysteresis mechanism is consistent with the proposed reaction path (Fig. 3) through examination of the potential dependence (and therefore dependence on the potential sweep direction) of the distinct surface species involved in the reaction. Specifically, we focus on the evolution of the Fe = O, Fe-OH, and Fe-OH···⁻OOH species. At low potentials, for example, $U = 0.6\,V_{RHE}$, the concentration of electrons at the surface is high (because the photoanode is donor-doped) and the recombination rate (step (2)) is fast, and therefore the surface is covered mostly by the long-lived Fe-OH and Fe-OH···⁻OOH species, whereas the short-lived Fe = O intermediates are minority species. Thus, the photocurrent at low potentials is dominated by the reaction path associated with the formation of Fe-OH···⁻OOH adions with increasing potentials, that is, in the $0.8\,V_{RHE} \leq U \leq 1.3\,V_{RHE}$ range, the electron concentration at the surface decreases due to band bending, thereby prolonging the lifetime of the Fe = O intermediates. Consequently, the surface concentration of Fe = O increases, leading to comparable contributions to the photocurrent from both the water and $H_2O_2$ photo-oxidation reactions. At high potentials, $U \geq 1.3\,V_{RHE}$, the surface recombination becomes negligible and the surface is covered mostly by long-lived Fe = O intermediates. If the potential is then swept in the opposite direction, that is from high to low potentials, the surface coverage of Fe = O decreases progressively. Unlike during the potential sweep from low to high potentials, the time scale that is required to form a significant coverage of Fe-OH···⁻OOH adions is much larger than in the reverse direction, consequently the $H_2O_2$ photo-oxidation reaction dominates. Thus, the different time scales that are associated with the evolution of different surface intermediates are manifested as hysteresis in the case of $H_2O_2$ photo-oxidation that involves concerted interaction of different intermediate specie.

## Discussion

The two-site interaction postulated here belongs to the LH class of surface reactions, where two different surface species mutually interact. This mechanism is inherently different from the single-site, multi-step process that accounts for the reported reaction mechanism of water photo-oxidation on haematite[24–26,34,35], as in ER reactions. Our two-site/LH reaction mechanism provides a favourable photo-oxidation path by splitting the charge transfer steps across two sites rather than the single-site/ER water photo-oxidation reaction, thereby helping to level the potential of the elementary steps involved in the reaction[20]. In a broader context, the nonlinear photocurrent voltammograms observed at certain $H_2O_2$ concentrations (see Fig. 1) correspond to a class of systems exhibiting negative differential resistance, typical of competing

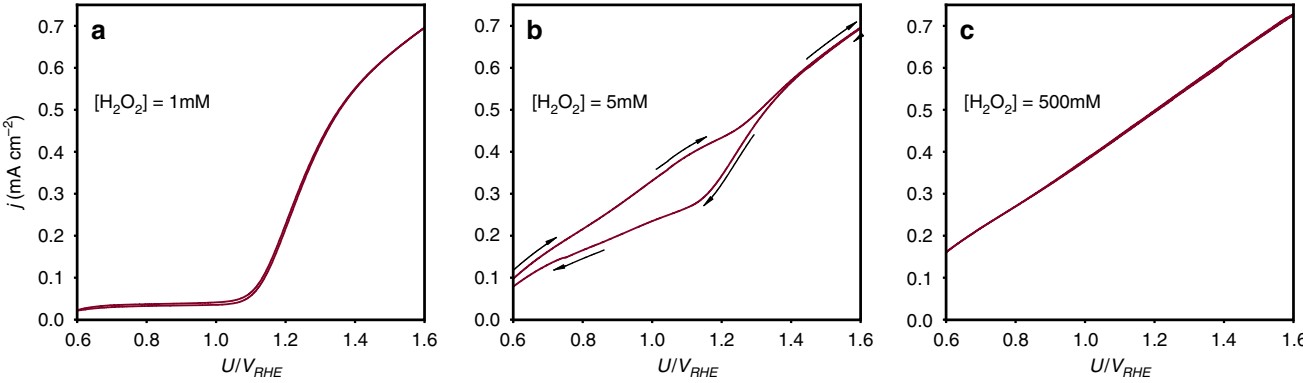

**Fig. 6** Experimental realization of hysteresis. Forth and back linear sweep voltammetry scans at different $H_2O_2$ concentrations in alkaline solution (1 M NaOH) under white LED illumination of 80 mW cm$^{-2}$: **a** 1 mM, **b** 5 mM and **c** 500 mM. Hysteresis is observed only at [$H_2O_2$]=5 mM. The arrows indicate the direction of potential sweep from low to high potentials and then back to low potentials

mechanisms with different time scales[47,48], as discussed above. Notably, the LH-like nature of the $H_2O_2$ photo-oxidation reaction bares similarity to catalytic CO oxidation[49], where bi-stability was observed and related to the generic character of LH reactions. This suggests that an appropriate spatiotemporal framework should be considered to account for concerted interaction between two sites, especially in cases where the simple single-site reaction scheme fails, as in the case of the $H_2O_2$ photo-oxidation reaction presented in this work.

It is noteworthy that two-site reactions on oxidized metal and metal oxide surfaces were previously suggested in the context of photocatalysis[50,51] and water photo-oxidation[20,32,52–56]. Particularly, hydroperoxide (HOO) adsorbed on metal sites participating in two-site LH-like reactions was suggested as part of the water oxidation mechanism on noble metals (Pt and Au) as well as $Co_3O_4$[36,57]. Oxo-bridged and peroxo-bridged Fe dimers ($Fe^{IV}-O-Fe^{IV}$ and $Fe^{IV}-O-O-Fe^{IV}$) were found on haematite surface in alkaline solution, but were reported not to participate in water oxidation[24,26]. The mechanism presented here, however, assumes the interaction of non-dimerized surface intermediates to form the O−O bond. The interaction of two non-dimerized surface intermediates in an LH-like mechanism has been suggested by Bockris[19] to take part in the water oxidation mechanism. Such a mechanism was indeed reported for water photo-oxidation on haematite at high light intensities and high potentials, where the photocurrent saturates[58]. The behaviour was attributed to conditions where the hole flux to the surface is high enough to allow the oxidation of nearest-neighbour sites, thereby reducing the surface recombination, consistent with our conclusion.

Given the ability of $H_2O_2$ to act as a hole scavenger, it often serves as a surrogate substrate that can be used to assess the effectiveness of co-catalysts for water photo-oxidation[29,30]. Thus, understanding the mechanism by which $H_2O_2$ extracts photo-generated holes may further provide insight into the effect of water oxidation co-catalysts on haematite photoanodes (and possibly other photoanode materials) and assist in rational design of better performing co-catalysts. In this context, it has been reported that sub-monolayer coverage of cobalt on haematite photoanodes has improved their water photo-oxidation performance, with no further enhancement as a result of additional co-catalyst loading[36]. Another study reported that the kinetic bottleneck that limits the water photo-oxidation performance of haematite photoanodes decorated with cobalt-phosphate (Co-Pi) water oxidation co-catalyst overlayers is effectively suppressed by sparse deposition on the haematite surface[37]. Trivial assignment of the detrimental effect observed with thicker layers to parasitic light absorption within the co-catalyst overlayer[59] was ruled out in this case by comparing back and front illumination measurements that were found to display similar trends[38]. These counter-intuitive observations cannot be accounted for by the "adaptive junction" mechanism that often prevails in thicker ion-permeable co-catalyst overlayers[60,61], or by alternative mechanisms suggested by other researchers, such as the formation of a Schottky-type heterojunction at the haematite/Co-Pi interface[62]. Thus, the mechanism by which ultrathin/sparse co-catalyst overlayers improve the water photo-oxidation performance of haematite photoanodes remains elusive[38,63,64]. The current work provides a possible insight to this enigma by considering that the water photo-oxidation reaction in the presence of sparse co-catalysts may proceed by a concerted interaction that involves two sites, one site from the photoanode and another site from the co-catalyst, similarly to the two-site $H_2O_2$ photo-oxidation reaction. Consistent with this view are the observations that ultrathin/sparse cobalt co-catalysts are more effective in enhancing water photo-oxidation at low potentials than at high potentials[36,37], indicating that the reaction proceeds via the co-catalyst at low potentials and via the haematite surface at high potentials[38]. This is similar to the effect of $H_2O_2$ reported in this work, suggesting a possible link between the respective photo-oxidation schemes. In view of these examples, we conclude that a LH-like mechanism in PEC reactions may be a general phenomenon and should be considered as a possible route for enhanced collection of photo-generated charge carriers.

To conclude, this work shows that the photo-oxidation mechanism of hydrogen peroxide ($H_2O_2$) on haematite photoanodes proceeds via a concerted interaction between two different surface species, one from the water photo-oxidation reaction and the other from $H_2O_2$ adsorption. This two-site LH-like reaction occurs in parallel with the single-site ER-like water photo-oxidation reaction. Splitting the photo-oxidation reaction into two sites instead of one facilitates the collection of photo-generated holes by the oxidized species by helping to level the potential of the elementary steps involved in the reaction, thereby reducing kinetic barriers. Moreover, we show that the competition between the $H_2O_2$ and water photo-oxidation reactions for holes, intermediate species and surface sites, involves processes with different time scales, thereby giving rise to bi-stability phenomena that may lead to negative differential resistance and hysteresis in the photocurrent voltammograms. Our findings suggest that the concerted interaction of two surface intermediates should be considered as a possible path towards improving the photoelectrode performance. This may shed light

on the effect of ultrathin/sparse co-catalysts for water oxidation on haematite as well as other photoanodes.

## Methods

**Haematite photoanode**. The samples consisted of a heteroepitaxial (110)-oriented Sn-doped (1%) haematite thin film photoanode (thickness ~30 nm) on Nb-doped $SnO_2$ (NTO) transparent electrode (thickness ~350 nm) deposited on an $a$-plane sapphire substrate. Details of the fabrication process and the microstructural characteristics of the photoanode can be found elsewhere[65].

**PEC measurements**. All measurements were conducted at ambient temperature in alkaline aqueous solutions (1 M NaOH in deionized water). For varying the $H_2O_2$ content, a 1 M NaOH + 0.5 M $H_2O_2$ solution was prepared in deionized water and diluted with 1 M NaOH in deionized water to reach the desired $H_2O_2$ concentrations. The pH of the electrolyte solution was kept fixed throughout the study. Linear sweep voltammetry measurements were carried out in a so-called cappuccino cell, as described in ref. [66] with the photoanode serving as the working electrode, Pt mesh as the counter electrode and an Hg/HgO reference electrode in 1 M NaOH aqueous solution. The reference electrode has a potential of 0.930 $V_{RHE}$. The measurements were carried out with a Zahner Zennium electrochemical workstation equipped with a CIMPS system[67]. The light source was a high power white LED (Zahner WLC02, 4300 K) with a maximum intensity of 120 mW cm$^{-2}$. The spectrum of this light source is shifted toward shorter wavelength than the AM1.5G solar spectrum, resulting in slightly higher photo-generation rate compared to proper solar simulators. To compensate for the spectral mismatch of our light source with respect to the AM1.5G solar spectrum, we reduced the light intensity to 80 mW cm$^{-2}$ instead of 100 mW cm$^{-2}$ in conventional solar simulators. Potential sweeps were performed by increasing the potential from 0.6 to 1.8 $V_{RHE}$ and subsequently decreasing it back to 0.6 $V_{RHE}$ with a sweep rate of 20 mV s$^{-1}$.

IMPS spectra were obtained by measuring both PEC impedance spectroscopy (PEIS) and intensity-modulated voltage spectroscopy (IMVS) at the corresponding operating point and then calculating the IMPS spectrum. This was done because of the better accuracy compared to direct IMPS measurement, as reported elsewhere[39]. The frequency was scanned from 10 kHz to 0.7 Hz and the perturbation amplitude was 10 mV for PEIS and 20 mV for IMVS.

**Spectroscopic characterization**. For Fourier transformed infrared spectroscopy (FTIR) measurements, samples were soaked for 30 min in a NaOH (1 M) aqueous solution or in a NaOH (1 M) + $H_2O_2$ (0.5 M) aqueous solution, then washed in DI water (18 MOhm) and dried in a gentle air flow. The FTIR spectra of the electrolyte solutions were recorded by a drop placed on an attenuated total reflection diamond crystal using a Nicolet 8700 FTIR spectrometer with a DTGS detector, with the background recorded on the clean diamond crystal. The spectra were recorded in the range from 4000 to 700 cm$^{-1}$, at 2 scanning resolution (0.24 cm$^{-1}$) and 36 scans. The FTIR spectra of the samples were recorded using an iN-10 FTIR microscope (Thermo Scientific) spectrometer equipped with a narrow-band liquid nitrogen-cooled MCTA detector. All single-beam spectra were measured against a background recorded from gold, with subtraction of the measured FTIR spectrum of the bare NTO/sapphire substrate. The spectra were recorded in the range from 4000 to 700 cm$^{-1}$, at 8 scanning resolution (1 cm$^{-1}$) and 128 scans; detection area was $100 \times 100$ µm$^2$. The FTIR data were collected using the OMNIC-Picta software.

**Model equations and numerical computations**. The rates of the reaction steps of the water and $H_2O_2$ photo-oxidation reactions, see eqs. (1)–(8), depend on the fractional surface coverage of species $x$, $\theta_x$, which are dimensionless by definition. The units of the electron and hole surface densities, $\sigma_e$ and $\sigma_h$, respectively, are given by their number surface density $[\sigma_e] = [\sigma_h] = $ mol m$^{-2}$. The units of the rate constants are given according to the corresponding reaction orders: $[k_1] = [k_2] = [k_3] = [k_4] = $ m$^5$ mol$^{-2}$ s$^{-1}$, $[k_5] = $ m$^3$ mol$^{-1}$ s$^{-1}$, and $[k_{-1}] = [k_6] = $ m$^2$ mol$^{-1}$ s$^{-1}$, and thus the units of other parameters read as mol m$^{-2}$ s$^{-1}$ for $p_1$, mol m$^{-2}$ for

$p_2$ and mol m$^{-3}$ for $p_3$ and $p_4$; $p_1$—the flux of photo-generated holes from the surface to the electrolyte; $p_2$—the electron density at the surface; $p_3$—the HOO$^-$ concentration in the electrolyte (which is directly proportional to the concentration of $H_2O_2$); and $p_4$—the OH$^-$ concentration in the electrolyte (related to the pH).

Introducing rescaled variables and parameters, $\hat{t} = t/\tau$, with $\tau = 1/p_4 k_5$, $\hat{p}_1 = p_1/k_6$, $\hat{p}_2 = p_2/\sigma_0$, with $\sigma_0 = k_6^{-1}\tau^{-1} = p_4 k_5/k_6$, $\hat{p}_3 = p_3/p_4$, $\hat{\sigma}_h = \sigma_h/\sigma_0$, $\hat{k}_1 = k_1 p_4/k_6$, $\hat{k}_2 = k_2 p_4/k_6$, $\hat{k}_3 = k_3 p_4/k_6$, $\hat{k}_4 = k_4 p_4/k_6$, $\hat{k}_{-1} = k_{-1}/k_6$, we obtain a combined set of dimensionless kinetic equations for reactions (1)–(8). For ease of reading the hat signs were removed in what follows and in Table 1:

$$\frac{\partial \theta_{Fe-OH}}{\partial t} = k_{-1} p_2 \theta_{Fe=O} + k_4 p_4 \theta_{Fe} \sigma_h + 2 k_6 \theta_{Fe=O} \theta_{Fe-OH\cdots OOH} \sigma_h \quad (12)$$

$$- k_1 p_4 \theta_{Fe-OH} \sigma_h - k_5 p_3 (\theta_{Fe-OH} - \theta_{Fe-OH\cdots OOH}),$$

$$\frac{\partial \theta_{Fe=O}}{\partial t} = k_1 p_4 \theta_{Fe-OH} \sigma_h - k_{-1} p_2 \theta_{Fe=O} - k_6 \theta_{Fe=O} \theta_{Fe-OH\cdots OOH} \sigma_h - k_2 p_4 \theta_{Fe=O} \sigma_h, \quad (13)$$

$$\frac{\partial \theta_{Fe-OOH}}{\partial t} = k_2 p_4 \theta_{Fe=O} \sigma_h - k_3 p_4 \theta_{Fe-OOH} \sigma_h, \quad (14)$$

$$\frac{\partial \theta_{Fe}}{\partial t} = k_3 p_4 \theta_{Fe-OOH} \sigma_h - k_4 p_4 \theta_{Fe} \sigma_h, \quad (15)$$

$$\frac{\partial \theta_{HOO}}{\partial t} = k_5 p_3 (\theta_{Fe-OH} - \theta_{Fe-OH\cdots OOH}) - k_6 \theta_{Fe=O} \theta_{Fe-OH\cdots OOH} \sigma_h. \quad (16)$$

We note that our model addresses the working conditions of haematite photoanodes at high pH levels[4–9].

For analysis, we first solve numerically the model equations under the charge and site conservation constraints described by Eqs. (9) and (10), respectively, and examine all the possible steady-state solutions (fix points) to Eqs. (12) through (16). The equations have been numerically solved by continuation method using the publically available package AUTO[68]. To determine the temporal stability of each steady-state solution, we linearize Eqs (12)–(16) about the fix points and solve the standard eigenvalue problem, while correlating the negative eigenvalues with stable solutions.

The dimensionless parameters $p_{1,2,3,4}$ for Eqs (12)–(16) are unknown, and thus the results are qualitative in nature. Consequently, we compute and compare only the qualitative trends. The dimensionless rate constants were chosen such that their ratios are in accord with reported values for water photo-oxidation on haematite[24]. The rate constants for the $H_2O_2$ photo-oxidation are rationalized in points (a) through (d) in the Results. The linear potential dependence of $p_1$ is chosen to be significantly smaller than 1, as most photo-generated holes do not arrive at the surface, see Table 1. The dimensionless OH$^-$ concentration, $p_4$, was set to be equal to 1. We cannot derive a simple relation between $\sigma_h$ and $p_2$, though such relation is expected from steps (1) and (2). The model parameters are summarized in Table 1. Additionally, we use a dimensionless potential scaled by $k_B T/q$ and also normalized, i.e., $\varphi = U/U_{sat}$, where $U_{sat} = 12$ is the minimal potential for which all photocurrents for different $H_2O_2$ concentrations are considered to be saturate, $k_B$ is Blotzmann constant, $T$ is temperature, and $q$ is elementary charge.

To test the validity of the LH-like mechanism proposed herein, a possible ER-like mechanism for $H_2O_2$ photo-oxidation was also considered. The alternative mechanism excludes reaction (7) and instead involves a direct interaction between HOO$^-$ and Fe = O, implying that $k_5 p_3 (\theta_{Fe-OH} - \theta_{Fe-OH\cdots OOH}) \rightarrow 0$ and $k_6 \theta_{Fe} = O \sigma_h \theta_{Fe-OH\cdots OOH} \rightarrow k_6 p_3 \sigma_h \theta_{Fe} = O$. The latter can then be combined with the linear $k_2 p_4 \sigma_h \theta_{Fe} = O$ term in Eq. (13), thereby removing the qualitative nonlinear nature of the solutions. Therefore, the resulting photocurrent based upon the ER-like mechanism fails to capture the non-monotonous behaviour observed in Fig. 1.

**Table 1 Summary of dimensionless control parameters and the rate constants used for the numerical computation of the results presented in Fig. 4**

| Control parameter | Value | Interpretation | Rate constant | Value |
|---|---|---|---|---|
| $p_1^c$ | 50 x exp(−4) | Hole flux to the electrolyte that is independent of the potential | $k_1$ | 0.16 |
| $p_1$ | 1.2 x $\varphi$ − 0.02 | Hole flux to the electrolyte that is dependent on the potential | $k_{-1}$ | 2 |
| $p_2$ | 50 x exp (−12 x $\varphi$) | Electron density at the surface | $k_2$ | 0.002 |
| $p_3$ | $\in [0, 0.5]$ | [HOO$^-$]~[$H_2O_2$] | $k_3$ | 0.002 |
| | | | $k_4$ | 0.002 |
| | | | $k_5$ | 1 |
| | | | $k_6$ | 1 |

## Data availability

All data generated or analysed during this study are included in this published article (and its Supplementary Information Files).

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

## Acknowledgements

This research was supported by the Ministry of National Infrastructures, Energy and Water Resources of Israel (grant no. 3-11430), the Ministry of Science and Technology of Israel (grant no. 3-14423), the European Research Council under the European Union's Seventh Framework programme (FP/200702013)/ERC (grant agreement no. 617516), and the Adelis Foundation. Y.Y.A. acknowledges support by the Kreitman Fellowship, B.G. acknowledges the support of the Blaustein Centre for Scientific Cooperation's postdoctoral fellowship and D.A.G. acknowledges support by Marie-Sklodowska-Curie Individual Fellowship no. 659491. The experiments were performed using central facilities at the Technion's Hydrogen Technologies Research Laboratory (HTRL) supported by the Adelis Foundation and the Solar Fuels I-CORE program of the Planning and Budgeting Committee and the Israel Science Foundation (grant no. 152/11). We thank Maytal Caspary Toroker for useful comments on the manuscript.

## Author contributions

A.Y. and H.D. conceived the idea. H.D., D.K., A.T., and B.G. designed and performed the experiments. S.K. performed the FTIR analysis. D.A.G. provided the haematite photo-anodes. Y.Y.A., H.D., I.V.-F., A.R., and A.Y. designed the chemical modelling. Y.Y.A. and A.Y. performed the theoretical analysis and wrote the first draft. A.Y., I.V.-F. and A.R. supervised the research. All co-authors contributed to the analysis and to the writing of the final manuscript.

## Additional information

**Competing interests:** The authors declare no competing interests.

