## [Peer Review File · Nature Communications]

Reviewers' comments:

Reviewer #1 (Remarks to the Author):

The authors have adequately addressed comments raised previously. The work should be published without further delays.

Reviewer #4 (Remarks to the Author):

The main conclusion of the paper is that the photo-oxidation mechanism of hydrogen peroxide on a hematite photoanode proceeds via a concerted interaction between HOO⁻ anion and Fe=O intermediate of water oxidation (eq. 2b). In response to my comment of the previous version regarding lack of spectroscopic evidence in support of the proposed mechanism, the authors have added infrared spectra (Fig. S1) that exhibit a frequency red shift of the broad OH stretch absorption in the 3200 cm⁻¹ region of the aqueous hematite sample and a shoulder between 3600-3700 cm⁻¹ when adding hydrogen peroxide, both of which are attributed to H₂O₂ adsorption on the Fe₂O₃ surface in the form of FeOH...-OOH-. Frequency shifts of OH stretch modes are not compelling for demonstrating specific interaction of HOO⁻ with surface Fe-OH groups, while measurement of OO stretch modes would be. As Nakato (Ref. 51) has demonstrated for the case of TiO₂ water oxidation catalysts, surface hydroperoxides can be selectively observed by infrared spectroscopy by monitoring the OO stretch mode in the 850 – 800 cm⁻¹ region, including positive identification of the mode by ¹⁸O isotope labeling (¹⁸O labeled hydrogen peroxide is commercially available). These are distinct from physisorbed H₂O₂, which absorbs at 877 cm⁻¹. How are alternative intermediates such as surface FeOOH groups and the (1-electron w.r.t H₂O₂) superoxide ruled out? The latter has recently been observed by steady state spectroelectrochemistry for hematite by Zhao (Ref. 55) and previously for Ni oxyhydroxide by Koper (Chem. Sci. 7, 2639 (216)). These examples reflect state of the art spectroscopic identification of proposed electrocatalytic intermediates at the metal oxide-aqueous interface. As the authors point out, some intermediates might be too short lived for spectroscopic detection lacking ultrafast time resolution, but this is highly speculative and does not justify lack of trying (spectroelectrochemical monitoring while varying the applied potential around oxidation onset (low overpotential) might allow monitoring of Fe=O sites interacting with FeOH...-OOH moieties on slow time scales).

For communicating a mechanistic study of wider implications to a scientifically broader audience, use of relevant state of the art physico-chemical methods is essential. This is not the case for this study. For a general chemistry audience or an electrochemical audience it may be different; it could be argued that spectroscopic studies are beyond the scope of the project. Communication of new mechanistic proposals without utilizing established methods which allow structural identification of proposed electrocatalytic intermediates, which may require collaboration with a spectroscopy group, is not suitable for a Nature subjournal.

Reviewer #5 (Remarks to the Author):

The authors investigate the photooxidation of H₂O₂ on hematite photoanodes as a function of H₂O₂ concentration, voltage and illumination intensities. They observe qualitative changes in the photocurrent-voltage relation with both the H₂O₂ concentration and the illumination intensity. These nonlinear dependencies are explained with a mean field model taking into account both water

oxidation and H₂O₂ oxidation. The heart of the model is a Langmuir-Hinshelwood mechanism for the formation of O₂ from H₂O₂ and the competition of water oxidation and H₂O₂ oxidation for the same surface intermediates. The model predicts bistable behavior which could also be detected in the experiments. I consider this an important result, being sufficiently interesting by itself to warrant publication in Nature communications. However, I have quite a large number of technical questions and concerns regarding the model. I only can recommend publication of the ms if the authors are able to satisfactorily resolve all of the points addressed in the following:

1) Eq. 3 is not correct in the form given. It should read $(\partial\sigma_h)/\partial t = p_1 - \sum_i(r_i) - sr$, where r_i denotes an individual reaction which is currently on the lhs of eq. (3), the sum goes over the five reactions, and sr is the surface recombination rate. The other symbols are as in the ms. (In words, the change of the hole density with time is the difference between the flux of holes from the semiconductor bulk to the surface and the reaction of holes due to chemical reactions or surface recombination with electrons.) Assuming that the first '=' sign should be a '-' sign, the surface recombination rate is missing. It is not clear to me why the authors can neglect it.

2) I do not understand why reaction (1b) should denote a surface recombination step as claimed in the ms (line 272 ff, '... minus the flux of holes consumed by the surface recombination reaction (step 1b). This step describes the reduction of FeO in alkaline solution with electrons (rather than by injection of a hole). In my understanding, step 1b involves neither surface recombination nor the flux of holes, if it did so, it should show up in eq. 3. – I also do not understand this asymmetry in the forward and backward reaction 1a/1b, which might be due to the fact that I am not sufficiently familiar with the literature on water photooxidation on hematite anodes, but some clarifying sentences would help a general readership to understand the mechanism.

3) The normalization of the current in Fig. 4 appears strange to me. Furthermore, it is somewhat misleading that the normalization does not coincide with the normalization done in Fig. 1, top right. In my understanding, in Fig. 1 right the current is normalized to the current at the respective potential, yielding to a normalized current for 0,5 M H₂O₂ that is 1 in the entire voltage range, whereas in the latter it is normalized to the current at U_{sat} , such that j/j_{500} decreases with potential. Is the quantity that is plotted in Fig. 4 measurable?

4) The proportionality constant of 'U proportional to ln (p₂)' is not given.

5) Why is the hole flux to the surface, p_1 , independent of U?

6) I understand that the absolute values of the rate constants k_i are not accessible and therefore the authors prefer to write the equations in terms of dimensionless quantities. However, how the model S1 to S5 is written suggests that the authors wrote the model in a dimensional form in the first place, making the transition to the dimensionless case simply by omitting all the units. The non-dimensionalization should be done in a proper way, defining dimensionless time and grouping the parameters such that a minimal number of parameters appears in the model. Otherwise, the authors should add units to their physical quantities. For example, in Table 1 p_1 and p_2 are put in relation to each other, however, since the hole flux to the surface (the interpretation of p_1) and the electron density at the surface (the interpretation of p_2) in their dimensional form have different units, the relation $p_2 \propto p_1$ remains obscure.

Response to Referee #4:

While we agree that rigorous spectro-electrochemistry is a valuable methodology to study surface specie, we disagree that it should be a prerequisite for publication. Our view is also echoed in the reports the other three referees, two of which had at first similar concerns and were eventually convinced otherwise with the additions of the *ex-situ* IR spectroscopy results in the previous version, e.g., referee #1 "*The authors have adequately addressed comments raised previously*".

The type of spectro-electrochemical analysis suggested by the referee is available only in a small number of highly specialized laboratories, and as such is not feasible for us within a reasonable time frame. The implication by the referee "*...which may require collaboration with a spectroscopy group*" also indicates that. On the other hand, the referee writes "*For communicating a mechanistic study of wider implications to a scientifically broader audience, use of relevant state of the art physico-chemical methods is essential.... For a general chemistry audience or an electrochemical audience it may be different; it could be argued that spectroscopic studies are beyond the scope of the project.*". We are glad to see that the referee appreciates our results as significant but evidently, if the referee agrees that our results are valid for a professional audience then there shouldn't be any more doubt for the broader audience. We genuinely respect his/her opinion about spectro-electrochemistry analysis. However, we and the other three referees do not feel that this opinion is a prerequisite and that it should preclude or delay publication. Notably, there have been outlined mechanisms similar to ours that have been published in high impact journals based solely on electrochemical results, e.g., see Fig. 8 in JACS **137**, 6629 (2015).

Furthermore, we would like to stress that the main conclusion of the paper is indeed "*that the photo-oxidation mechanism of hydrogen peroxide on a hematite photoanode proceeds via a concerted interaction*" of two surface species, whose exact nature is suggested but not completely proven. The surface specie may be "*HOO⁻ adion and Fe=O intermediate of water oxidation*", but they may also be other specie and this would not drastically change the main conclusions of this work. A different composition of these surface species will not change the broader conclusions of this work based on the reaction rate analysis. To prevent further misunderstanding we highlight this in the manuscript, the following text modifications were added in the revised manuscript (marked by green font color):

(Results and Discussion, p. 11 lines 261-268) "*It is noted that the exact molecular identity of the surface specie involved in the reaction is beyond the scope of this work, and it remains to be verified by spectroelectrochemical studies.⁴⁵ However, this specific detail is not critical to the modelling results that follow, and alternative surface specie may be considered without affecting the qualitative results of the analysis. The salient point is that the H₂O₂ photo-oxidation reaction mechanism involves a concerted interaction of two surface sites, as in LH reactions, consuming two holes and yielding an oxygen molecule.*"

(Conclusions, p. 17 lines 420-424) "*This work shows that the photo-oxidation mechanism of hydrogen peroxide (H₂O₂) on a model system hematite photoanode proceeds via a concerted interaction between*

two different surface specie, one from the water photo-oxidation reaction and the other from H_2O_2 adsorption. This two-site LH-like reaction occurs in parallel with the single-site ER-like water photo-oxidation reaction."

We hope that the referee will take this perspective into a consideration and reconsider his/her recommendation.

Response to Referee #5:

We are grateful to the referee for his/her careful/constructive report and positive recommendation. We have addressed all comments and reply by the same order as they appear in the report. We hope that the referee will find the changes (marked by font green color in the manuscript) satisfactory and warrant publication.

1) "Eq. 3 is not correct in the form given. It should read $(\partial\sigma_h)/\partial t = p_1 - \sum_i(r_i) - sr$, where r_i denotes an individual reaction which is currently on the lhs of eq. (3), the sum goes over the five reactions, and sr is the surface recombination rate. The other symbols are as in the ms. (In words, the change of the hole density with time is the difference between the flux of holes from the semiconductor bulk to the surface and the reaction of holes due to chemical reactions or surface recombination with electrons.) Assuming that the first '=' sign should be a '-' sign, the surface recombination rate is missing. It is not clear to me why the authors can neglect it."

We thank the referee for highlighting that the explanation of Equation 3 was confusing. We have revised the paragraph related to equation 3 to clarify the concerned issue. The revised related text new reads: (p. 11 lines 270-279): "To uncover the nonlinear nature of the elementary steps in the water and H_2O_2 photo-oxidation reactions that result in the measured non-monotonic photocurrents for certain H_2O_2 concentrations (see Figure 1), we derive kinetic equations (see SI for details) and supplement them by the hole flux from the surface to the electrolyte that is given by the sum of the forward chemical reactions (1a), (1c-e), and (2c):

$$k_1 p_4 \theta_{Fe-OH} \sigma_h + k_2 p_4 \theta_{Fe=O} \sigma_h + k_3 p_4 \theta_{Fe-OOH} \sigma_h + k_4 p_4 \theta_{Fe} \sigma_h + k_6 \theta_{Fe-OH} \theta_{OOH} \theta_{Fe=O} \sigma_h = p_1 \quad (3)$$

where p_1 is the hole flux from the surface to the electrolyte,⁴⁶ which in general depends on the illumination intensity and potential, and p_4 is the OH^- concentration in the electrolyte, which depends on the electrolyte composition), θ_x is the fractional surface coverage of species x , and σ_h corresponds to holes of reacting sites. It is useful to regard p_1 as the charge conservation constraint.⁴⁶"

We note that the inverse hole flux due to surface recombination is not included in eq. 3, as it accounted for only in eq. 5 (see also response to comment 2). For discussion of the effect of bulk recombination, please see the response to comment 3.

2) "I do not understand why reaction (1b) should denote a surface recombination step as claimed in the ms (line 272 ff, '... minus the flux of holes consumed by the surface recombination reaction (step 1b). This step describes the reduction of FeO in alkaline solution with electrons (rather than by injection of a hole). In my understanding, step 1b involves neither surface recombination nor the flux of holes, if it did so, it should show up in eq. 3. – I also do not understand this asymmetry in the forward and backward reaction 1a/1b, which might be due to the fact that I am not sufficiently familiar with the literature on water photooxidation on hematite anodes, but some clarifying sentences would help a general readership to understand the mechanism."

This issue indeed follows the rationale developed in this field, e.g., Zandi et al. suggest that FeO is a surface intermediate that participates in both forward and backward reactions. Reaction 1b transfers an electron into the electrolyte and thus effectively reduces the measured net photo-current calculated from the difference between the hole flux from the surface to the electrolyte (p_1 , eq. 3) and the inverse hole flux due to reaction 1b. The calculation of the (normalized) photocurrent in eq. 5 is based on that concept. As suggested by the reviewer, the following clarifying sentence was added in p. 9 line 221-223: "When both reactions 1a and 1b occur, their combination is effectively a recombination step."

3) "The normalization of the current in Fig. 4 appears strange to me. Furthermore, it is somewhat misleading that the normalization does not coincide with the normalization done in Fig. 1, top right. In my understanding, in Fig. 1 right the current is normalized to the current at the respective potential, yielding to a normalized current for 0,5 M H₂O₂ that is 1 in the entire voltage range, whereas in the latter it is normalized to the current at U_{sat} , such that j/j_{500} decreases with potential. Is the quantity that is plotted in Fig. 4 measurable?"

We are grateful to the reviewer for pointing out that this issue requires further clarification. We have modified both Figure 4 and the respective text to address the agreement. The following text was added on p. 12 lines 281-296: "Next we define the photocurrent that is related to the flux of holes across the surface (p_1) minus the flux of holes consumed by the surface recombination reaction (step (1b))⁴⁶ :

$$\frac{j}{j_{max}} = 1 - \frac{k_{-1}\theta_{Fe=0}p_2}{p_1} \quad (5)$$

We emphasize that this normalized photocurrent regards only the reaction kinetics at the surface and it does not account for bulk processes that control the hole flux to the surface. From experiments it is evident that p_1 is often potential dependent, but the dependence is relatively weak. For instance, in the results presented in Figure 1 the dependence is essentially linear (see j_{500} in Figure 1). In contrast to p_1 , p_2 displays strong dependence on the applied potential, $p_2 \propto \exp(U/U_{sat})$ where U_{sat} is the minimal potential for which all photocurrents are considered to be in the saturation for different H₂O₂ concentrations (see SI for details). Consequently, j/j_{max} in Equation (5) includes an empirical normalization by p_1 (similarly to the normalization by j_{500} in Figure 1 top right panel). Notably, the shape of the normalized photocurrent is essentially robust with respect to the potential dependence of p_1 , as shown here by comparing Figures 4(a) and (b) in which p_1 is taken to be linearly proportional to U or independent of U , respectively. This implies a generic mechanism that is hidden in this physicochemical process that we unfold, in what follows, by using a bifurcation theory via keeping p_1 as constant for simplified analysis purposes, $p_1 = p_1^c$."

4) "The proportionality constant of 'U proportional to ln (p2)' is not given."

The proportionality constant is $k_B T/q$. We have included this missing detail in the SI, see lines 57-63.

5) "Why is the hole flux to the surface, p_1 , independent of U ?"

The referee is correct and indeed p_1 is dependent on the potential, from Figure 1 it is evident to be of linear form (see j_{500}). However, this dependence is effectively negligible (as we now show) and for the sake of generality via bifurcation analysis, we treat it as constant (as the referee probably expects). To clarify this issue also for a general audience, we have modified Figure 4 with a plot showing potential dependence, panel (a), and respectively clarified the text on p. 12, see reply to point 3 above.

6) "I understand that the absolute values of the rate constants k_i are not accessible and therefore the authors prefer to write the equations in terms of dimensionless quantities. However, how the model S1 to S5 is written suggests that the authors wrote the model in a dimensional form in the first place, making the transition to the dimensionless case simply by omitting all the units. The non-dimensionalization should be done in a proper way, defining dimensionless time and grouping the parameters such that a minimal number of parameters appears in the model. Otherwise, the authors should add units to their physical quantities. For example, in Table 1 p_1 and p_2 are put in relation to each other, however, since the hole flux to the surface (the interpretation of p_1) and the electron density at the surface (the interpretation of p_2) in their dimensional form have different units, the relation $p_2 \ll p_1$ remains obscure."

The referee is correct about his/her impression, in fact our rationale was to remove technical jargon for simplicity but we agree that it may not be suitable now for Nature Communication. We have revised the SI respectively, to include the missing details in p. 1-2 lines 22-37: "The rates of the reaction steps of the water and H_2O_2 photo-oxidation reactions, see eqs. (1) and (2), are dependent on the fractional surface coverage of species x , θ_x , which are dimensionless by definition. The units of the electron and hole surface densities, σ_e and σ_h , respectively, are given by their number surface density $[\theta_x]=[\sigma_e]=[\sigma_h]=mol \cdot m^{-2}$. The rate constants are given in the following units, according to the corresponding reaction orders: $[k_1] = [k_2] = [k_3] = [k_4] = m^5 \cdot mol^{-2} \cdot sec^{-1}$, $[k_5] = m^3 \cdot mol^{-1} \cdot sec^{-1}$, and $[k_{-1}] = [k_6] = m^2 \cdot mol^{-1} \cdot sec^{-1}$, and thus the units of other parameters read as $mol \cdot m^{-2} \cdot sec^{-1}$ for p_1 , $mol \cdot m^{-2}$ for p_2 , and $mol \cdot m^{-3}$ for p_3 and p_4 ; p_1 – the flux of photo-generated holes from the surface to the electrolyte; p_2 - the electron density at the surface; p_3 - the HOO^- concentration in the electrolyte (which is directly proportional to the concentration of H_2O_2); and p_4 - the OH^- concentration in the electrolyte (related to the pH).

Introducing rescaled variables and parameters, $\hat{t} = t/\tau$, with $\tau = 1/p_4 k_5$, $\hat{p}_1 = p_1/k_6$, $\hat{p}_2 = p_2/\sigma_0$, with $\sigma_0 = k_6^{-1} \tau^{-1} = p_4 k_5/k_6$, $\hat{p}_3 = p_3/p_4$, $\hat{\sigma}_h = \sigma_h/\sigma_0$, $\hat{k}_1 = k_1 p_4/k_6$, $\hat{k}_2 = k_2 p_4/k_6$, $\hat{k}_3 = k_3 p_4/k_6$, $\hat{k}_4 = k_4 p_4/k_6$, $\hat{k}_{-1} = k_{-1}/k_6$, we obtain a combined set of dimensionless kinetic equations for reactions (1) and (2). For ease of reading the "hut" signs were removed in the following dimensionless kinetic equations and in Table 1:"